# Femtosecond Er-Doped All-Fiber Laser with High-Density Well-Aligned Carbon-Nanotube-Based Thin-Film Saturable Absorber

**DOI:** 10.3390/nano12213864

**Published:** 2022-11-02

**Authors:** Dmitriy A. Dvoretskiy, Stanislav G. Sazonkin, Ilya O. Orekhov, Igor S. Kudelin, Lev K. Denisov, Valeriy E. Karasik, Viatcheslav N. Agafonov, Valery N. Khabashesku, Valeriy A. Davydov

**Affiliations:** 1Scientific and Educational Center “Photonics and IR Technology”, Bauman Moscow State Technical University, 105005 Moscow, Russia; 2Department of Physics, University of Colorado, Boulder, CO 80309, USA; 3GREMAN, CNRS, UMR 7347, INSA CVL, Université de Tours, 37200 Tours, France; 4Department of Materials Science and Nanoengineering, Rice University, Houston, TX 77005, USA; 5L.F. Vereshchagin Institute for High Pressure Physics, Russian Academy of Sciences, 108840 Moscow, Russia

**Keywords:** erbium-doped fiber lasers, carbon nanotubes, ultrashort pulse generation, nonlinear polarization evolution effect, hybrid mode-locking

## Abstract

We have studied the ultrafast saturation behavior of a high-density well-aligned single-walled carbon nanotubes saturable absorber (HDWA-SWCNT SA), obtained by a high-pressure and high-temperature treatment of commercially available single-wall carbon nanotubes (SWCNTs) and related it to femtosecond erbium-doped fiber laser performance. We have observed the polarization dependence of a nonlinear optical saturation, along with a low saturation energy level of <1 fJ, limited to the detector threshold used, and the ultrafast response time of <250 fs, while the modulation depth was approximately 12%. We have obtained the generation of ultrashort stretched pulses with a low mode-locking launching threshold of ~100 mW and an average output power of 12.5 mW in an erbium-doped ring laser with the hybrid mode-locking of a VDVA-SWNT SA in combination with the effects of nonlinear polarization evolution. Dechirped pulses with a duration of 180 fs were generated, with a repetition rate of about 42.22 MHz. The average output power standard deviation was about 0.06% RMS during 3 h of measurement.

## 1. Introduction

Potential implementations of nanoscale carbon forms in nonlinear optics are among the most urgent problems currently facing modern materials science and quantum-physical instrumentation development [1]. For example, one of the most important elements of a femtosecond mode-locked (ML) fiber laser is a saturable absorber based on nanomaterials—single-walled carbon nanotubes (SWCNTs). In 2003, this type of saturable absorber was used in mode-locked lasers for the first time [2]. At present, the effective generation of ultrashort pulses (USPs) has been implemented using various ML techniques [3]. Based on semiconductor materials, there are various types of saturable absorbers, such as SESAM [4], topological insulators [5], black phosphorus-based saturable absorbers (SA) and their analogs [6], bismuthene SA [7], ReSe_2_ [8], MoS_2_ [9], SnS_2_ [10]. SAs and other 2D van der Waals materials [11,12] have ultrafast saturable absorption at the infrared spectral region. Among these materials, a special place is occupied by reliable and popular saturable absorbers based on various carbon nanostructures, such as single-wall carbon nanotubes, graphene, and graphene oxide [13,14,15]. In addition to the cost advantages, SWCNTs-SAs significantly improve the noise characteristics of lasers and have an ultrafast relaxation time [16,17,18]. Moreover, SWCNTs can be easily introduced as a doping element into polymer films, with high optical quality in the near-infrared spectral region, ensuring the use of this saturable absorber as a mode-locker in the all-fiber resonator setup and providing an advantage for laser tuning [19,20,21]. It should be noted that SWCNTs can operate in a wide spectral range (up to 500 nm or more) due to the presence of dispersion in the nanotube diameters and their chirality during the manufacturing process [13,19,20,22,23,24]. Even with the well-established production technology of the SWCNTs themselves, there is an issue regarding controlling the response time of the SWCNTs-SA during the film manufacturing process, which is critical for ML. In particular, it is known that the response time of SWCNTs is determined by the complex intratubular interaction (the excited state energy is transmitted to metal tubes and then quenched) [16]; hence, the response time depends on the SWCNTs’ production technology [25]. Moreover, an increase in SWCNTs concentration in the polymer films does not lead to an improvement in modulation depth, but rather leads to an increase in unsaturated losses [26,27]. Furthermore, the SWCNTs orientations are randomly distributed, which makes it impossible to control the polarization dependence of the SA. Immediately after fabrication, traditional carbon nanotubes are a disordered structure of filaments intertwined with each other. Such a disordered structure significantly affects the entire range of the optical characteristics of SA, which, in turn, makes it difficult to obtain well-reproducible generation modes in USP lasers. [28]. On the other hand, the presence of a polarization dependence in a saturable absorber would make it possible to reduce the variability of generation regimes during laser assembly. It would also make it possible to more efficiently single out the low-intensity direction of the polarization ellipse in the fiber lasers, which would simplify the self-starting of the mode locking process.

In previous works, it has been shown that these problems can be solved by ordering the structure of carbon nanotubes [29,30]. To date, several research groups have developed techniques for synthesizing ordered carbon nanotubes using a surface modification of the substrates, hydrophobic films, a liquid–air interface, and catalysts [31,32,33,34], as well as growing crystals with vacancies in the form of SWCNTs [30] and the use of a plasma-enhanced chemical vapor deposition technique [35]. However, even given the great progress in the study of nanostructures, the production of such materials, with the required parameters and in large quantities, is a difficult task. In the present work, the ultrafast nonlinear optical properties of a saturable absorber obtained by the high-pressure and high-temperature treatment of commercially available SWCNTs were investigated. It should be noted that the method described in this work is much simpler and cheaper than other methods of obtaining ordered carbon nanostructures. In addition, treatment with high pressure and high temperature allows for the obtaining of multilayer ordered carbon structures of large density and in relatively large quantities. We have shown that there is a significant impact on the ultrafast optical properties of SA due to the well-aligned and high-density structure of the newly developed SWCNTs, which is related to the characteristics of the ML laser.

## 2. Production and Nonlinear Optical Properties of an HDWA-SWCNTs SA

The investigation of the transformation processes of single- and multi-walled carbon nanotubes and C60-pipodes, under conditions of hydrostatic and non-hydrostatic compression, have revealed the existence of the orientational ordering effect in the cases of distinct non-hydrostatic compression [36,37,38]. The orienting effect is associated with the presence of inhomogeneous tensile stresses that arise in high-pressure apparatuses. Particularly, the clear ordering effect of non-hydrostatic compression on the nanotubes bundles has been seen in the case of SWCNTs.

High-density, well-aligned single-walled carbon nanotubes (HDWA-SWCNT) were obtained using a high pressure of 8 GPa and a high temperature of 400 °C applied to commercially available SWCNTs (HiPco™ Single-Walled Carbon Nanotubes, Atom Optoelectronics, Los Angeles, CA, USA). As can be seen in Figure 1, HDWA-SWCNTs have a pronounced orientation compared to their initial state. After pressure and temperature treatment, HDWA-SWCNTs were mixed with an aliphatic epoxy resin, which possesses optical transparency in the infrared region of the spectrum. Films with a thickness of 1 µm were obtained by cutting a polymerized mixture of epoxy resin with HDWA-SWCNTs. Later, these films were installed in a module with a saturable absorber, including nanotubes (SAINT module) formed by two FC/APC connectors, for further study.

Figure 2a shows Raman spectrums of initial SWCNTs and HDWA-SWCNTs obtained with an unpolarized pump at a wavelength of 641.4 nm. The Raman spectrums clearly show the presence of SWCNTs after the treatment due to the strong Raman mode at 1590.2 cm^−1^ [19,39]. On the other hand, the high pressure and temperature treatment, which should introduce nanotubes defects, resulted in significant increase at the 1324.9 cm^−1^ band [40]. HDWA-SWCNTs remained after treatment, with a diameter ≈ 1 nm (corresponding to the 225.7 cm^−1^ breathing mode of Raman spectra) [39], suitable for the the promotion of the successful mode-locking process. Figure 2b shows a small signal absorption spectrum of 1 µm thickness of a thin film. Note that SWCNTs absorb more than *l*_0_ ≈ 80% of the halogen lamp optical power in a wide spectral region of 1000–1650 nm.

Figure 3 shows the experimental setup on which the ultrafast saturation measurement was performed. For measurements, we used homemade ML erbium-doped fiber lasers with a pulse duration of 180 fs and 300 fs and a pulse repetition rate of 42.2 MHz and 11.1 MHz, respectively [41,42]. The laser radiation was directed to the splitter, where it was divided into two optical channels at a ratio of 50/50, preliminarily passing through a tunable attenuator. The radiation in one port of the splitter was sent directly to the radiation receiver, and a polarization controller in the SAINT module with the HDWA-SWCNTs under study were installed on the second port.

To satisfy the condition of a slowly saturating absorber, we sought to use the shortest probing pulses possible. Therefore, the duration of the laser pulse must be much shorter than the recovery time of the SA. Due to this assumption, the optical loss in the SA, depending on the energy of the input pulse *E_p_*, can be expressed as [43]:(1)$$l=l_{ns}+\Delta T\cdot \frac{1-\exp\left(-E_{p}/E_{sat}\right)}{E_{p}/E_{sat}}$$

Here, *l_ns_* represents non-saturable losses, Δ*T* shows the modulation depth of the losses (transmission), and *E_sat_* shows the energy saturation that accounts for a saturation behavior in the case of a slow saturable absorber.

Figure 4a shows the measured loss function of an HDWA-SWCNT thin film absorber compared to a similar absorber based on boron nitride doped carbon nanotubes using either 180 or 300 fs full width half maximum (FWHM) pulses. We observed a polarization dependence (up to a 3% difference in optical losses for 2 orthogonal polarizations) of optical saturation, which is direct evidence of the ordered structure of HDWA-SWCNT, as well as a low level of saturation energy *E_sat_* of <1 fJ, limited to the detector threshold used, while the maximal modulation depth was Δ*T* ≈ 12%. Note that the power meter’s minimal detecting optical power was ≈300 pW (model MTP 6000, IIT BSIIR, Minsk, Belarus). We have also varied the ultrashort pulse durations (see Figure 4b) in the wide range of 0.25–28 ps (at the FWHM pulse intensity level) by controlling the length of the output fiber SMF-28 using a homemade ultrashort pulse fiber laser [41]. Moreover, carbon nanotubes, doped with boron nitride, serve as an ideal platform for creating laser saturated absorbers, as well as a short response time, from a high modulation depth [28,42,44]. However, compared to C:BNNT, high-density SWCNTs films have the advantage of a much lower level of saturation energy and higher resistance to radiation damage. It is also worth noting that during the experiments, we observed a dependence of the modulation depth on the duration of the irradiating pulses, which proves a shorter response time (*t_resp_*) for HDWA-SWCNTs compared to boron nitride-doped nanotubes. It should be mentioned that the observed response time of HDWA-SWCNTs shows good agreement with the ultrafast carrier dynamic measurements in SWCNTs obtained by the pump–probe method [45].

Table 1 shows significant changes in the key nonlinear optical characteristics of conventional SWCNTs [28] and C:BNNT compared to HDWA-SWCNTs. HDWA-SWCNTs benefit from an ultrashort relaxation time *τ_resp_* < 250 fs, which is one of the best results in the class of the SWCNTs SA [46]. This obstacle, on one hand, provides reliable ML launching. On other hand, the low-saturation-energy level of HDWA-SWCNTs SA dramatically impacts ML laser performance, i.e., as an inefficient ultrashort pulse filter, it may ruin the ML operation [47]. In this way, the application of the HDWA-SWCNTs as an SA requires an additional mechanism of ML support in the laser resonator, such as the hybrid ML mechanism [3]. It should be noted that the low saturation energy level in the wavelength range of 1.3 to 1.7 μm, combined with a response time of about 250 fs and a wide absorption spectrum, makes these nanotubes attractive for use as an ultrafast and sensitive photodetector [48,49,50].

## 3. Mode-Locking by the HDWA-SWCNTs SA

Figure 5a shows the experimental setup of a femtosecond hybrid ML all-fiber laser. The HDWA-SWCNTs film was inserted into the SAINT module, formed by two FC/APC connectors as a slow passive SA. A fast mode-locking mechanism based on nonlinear polarization evolution (NPE) was provided by a commercially available isolator-polarizer (ISO-PM). The ISO-PM is used for two purposes. First, it blocks the back propagation of radiation in the resonator. Second, it weakens the orthogonal polarization in the ellipse, causing the radiation to be linearly polarized. In the process of pulse propagation, it is assumed that the state of the laser resonator changes nonlinearly under the action of phase and cross modulations, which arise due to the nonlinearity of the fibers, creating a phase shift between the orthogonally polarized components. This leads to a rotation of the polarization ellipse, but not to a change in its shape. Thus, upon multiple passages through the ISO-PM, the low-intensity polarization component is so weakened that this allows the mode-locking process to be triggered. Two polarization controllers (PCs) were used to adjust the mode-locking regimes. The resonator consists of two types of optical fibers, including erbium-doped fiber with positive group velocity dispersion (GVD) at 1550 nm and SMF-28 (manufactured by Corning Corp., Glendale, AZ, USA) with a negative GVD at 1550 nm. It should be noted that the length of the erbium-doped fiber is a 1.34 m, with a small signal core absorption of ~43 dB/m at the 980 nm pump wavelength and a dispersion parameter (D) of ~−30.7 ps/(nm·km) at 1550 nm, but the length of the SMF-28 (manufactured by Corning Corp.) is 3.6 m. The pump source for the erbium-doped fiber is used as a pigtailed single-mode laser diode at 980 nm. A 50/50 coupler is used for the output of laser radiation from the cavity.

We have realized a stable, single-pulse, self-starting USP generation through both PCs adjustments. The inset of Figure 5b shows the output pulse trace for the investigated USP generation (using the Infinium MSO9254A oscilloscope, Keysight Technologies, Santa Rosa, CA, USA) at a repetition rate of 42.2 MHz, corresponding to a total cavity length of ~4.93 m.

The ML threshold (see Figure 5b) was observed at an average pump power of ~104 mW. Moreover, we have not observed any pulse breakage by increasing the pump power up to 330 mW. It is worth noting that regarding such a low ML threshold, compared with NPE-based only, ML [41] is attributable to the applied hybrid ML technique in which the slow SA, represented by HDWA-SWCNTs SA, reliably triggers ML at a low pump power, while the pulse shortening is caused by the NPE effect [51]. It should be noted that the ML laser threshold corresponds well with the results of the PML (presented in Table 1) obtained by the ML technique based only on SWCNTs SAs for the significant value of unsaturated losses *l_ns_*. This fact proves that the observed USP is generated mainly by the action of the HDWA-SWCNTs SA.

As experimental confirmation of the statement that the key component providing the mode-locking launching is the SA based on the HDWA-SWCNTs, we present the measured data from the implemented Q-switching operation mode (also obtained by PCs adjustment), which naturally coexists in the laser resonator along with the ML operation, ensured by the slow relaxation time of the SWCNTs [52]. As can be seen from the inset to Figure 6, the spectral width of the output spectrum is about 4 nm (at FWHM). In this case, the pulse repetition rate estimated through a typical pulse train corresponded to 108 kHz, which is confirmed by the measured RF spectrum in the range from 80 kHz to 1 MHz.

Figure 7a shows the pulse spectra obtained for the USP regime at a maximum average output power of ~12.5 mW. The Gaussian-type spectrum has an FWHM of ~30 nm. Figure 8a shows the corresponding intensity autocorrelation traces and its Gaussian fitting, along with the pulse phase of the dechirped USP obtained at the laser output with a 2.75 m-long SMF-28 fiber from the coupler to an intensity autocorrelator (obtained from Swamp Optics LLC, GRENOUILLE Model 15-40-USB, Atlanta, GA, USA). Note that the Gaussian spectrum and pulse shape are inherent in stretched-pulse generation [53]. An estimation of pulse widths gives a value of 180 fs at FWHM. The calculated time-bandwidth product (TBP) for the Gaussian pulses is TBP ~0.67 (the typical TBP value for Gaussian pulses is ~0.441). Thus, the obtained Gaussian pulse is close to its bandwidth limit.

Figure 8b shows a pulse-width evolution at the laser output. In the experiment, a dependence was obtained that is quite accurately consistent with the function responsible for the Gaussian pulse evolution $\tau_{p}=\tau_{\min}\sqrt{1+{\left((L-L_{0})/L_{D}\right)}^{2}}$, with the dispersion length (*L_D_*) of (48.0 ± 1.0) cm, where *τ*_min_ is the minimum pulse duration, *L* is the length of the current output fiber, and *L*_0_ is the length of fiber when *τ*_min_. At the output, the SMF length *L_SMF_* ~2.75 m exhibited the minimum pulse duration position in the laser cavity for conservative dynamics of the Gaussian pulse evolution [54]. Therefore, the compression of pulses at the laser output is mainly governed by fiber dispersion. Moreover, the symmetrical shape of a second harmonic generation (SHG) FROG trace of the USP (Figure 7b) and the pulse-phase value close to zero (see Figure 8a) confirms the generation of the dechirped bandwidth-limited USP.

To characterize the short-term stability of the obtained USP generation, we measured the radio-frequency (RF) spectrum at the fundamental oscillation frequency of 42.2 MHz, depicted in Figure 9a (using a FEMTO 200 MHZ HCA-S-200M InGaAs photoreceiver and the ESA FSL 3 model.03; Rohde & Schwarz GmbH & Co. KG, Munich, Germany), with a signal-to-noise ratio (SNR) ~64 dB (with the ESA resolution of 300 Hz). The inset of Figure 9a shows the RF spectrum in the frequency range of 30 kHz–200 MHz (3-kHz resolution bandwidth). The high SNR ratio at the fundamental frequency and the absence of any Q-switch sidebands in the RF spectrum prove the pulse-to-pulse stability of the ML regime.

Moreover, we have measured the relative intensity noise (RIN) of the fiber laser with the maximum value of <−135.4 dBc/Hz, as shown in Figure 9b, in the range 30 Hz–100 kHz (with 250 Hz resolution using ESA SR770FFT, manufactured by Stanford Research Systems) and in the range 30 kHz–1000 kHz (with 300 Hz resolution using ESA manufactured by Rohde & Schwarz). As a result, we have obtained an RIN value for the ultrashort pulse fiber laser, mode-locked by HDWA-SWCNTs SA in a coaction with an NPE effect, that is comparable to that obtained previously by low-energy stable femtosecond optical combs [55].

At last, to understand the long-term stability of the obtained ML generation regime, we measured the output average power stability at the nearly maximum available output average optical power of ~11.3 mW and the Allan deviation of the repetition rate (for 5 Hz data using the 53132A manufactured by Agilent Inc) of the free-running fiber laser. Figure 10 shows the stability of the average optical output power with a standard deviation of ~0.06% RMS over a time of 3 h (measured by a PM200 power meter with an InGaAs S145C detector manufactured by Thorlabs Inc., Bergkirchen, Germany). As can be seen from the inset in Figure 10, the developed laser also has a relatively low pulse repetition rate deviation, with a value of 6.3 × 10^−8^ for an interval of 1 s, which is only determined by the temperature drift during the average time interval of 1 × 10^2^ s and environmental action. It should be noted that during the experiment, which lasted for several hours, we did not observe any pulse breakage or instability in the laser operation.

## 4. Conclusions

We have carefully investigated the optical properties of an ultrafast saturable absorber based on highly oriented high-density single-walled carbon nanotubes and related it to the obtained output characteristics of an erbium-doped femtosecond all-fiber laser. We have observed polarization dependence of a nonlinear optical saturation, proving the aligned structure of the SWCNTs, along with an ultra-low saturation energy level (*E_sat_* < 1 fJ limited to the threshold of the detector used) compared to conventional SWCNTs, while the modulation depth was ≈12%. Moreover, the observed ultrafast saturation behavior, with a response time of <250 fs, ensured a robust mode-locking launching in a hybrid erbium-doped ML fiber laser, with an ultra-low threshold at about 100 mW. We have demonstrated the generation of stable dechirped 180 fs stretched pulses at a central wavelength of 1555 nm with the 12.5 mW average output power corresponding to the <1.3 kW maximum peak power and <0.3 nJ maximum pulse energy, with an RIN value < −135.4 dBc/Hz (30 Hz–1000 kHz), at a 42.2 MHz repetition rate, with an SNR ratio ≈ 64 dB (with the resolution of ~300 Hz) in the erbium-doped ring laser hybrid mode-locked by HDWA-SWCNTs SA in co-action with a nonlinear polarization evolution effect. In this case, the Allan deviation of the repetition rate does not exceed ~3 × 10^−6^ at an average time of ~100 s, and the standard deviation of the average output power does not exceed ~0.06% RMS for 3 h of measurements. It should be noted that the low intensity noise in a fiber laser with a cavity length of more than 5 m was achieved by using a natural saturable absorber based on nanotubes in the laser. At the same time, the NEP effectively reduced the pulse duration by operating in the mode of a fast saturable absorber.

In summary, newly-developed HDWA-SWCNTs thin film SA demonstrates a low saturation energy level at 1.5 µm, along with an ultrafast response time and a wide absorption spectrum. The unique nonlinear optical properties of HDWA-SWCNTs make this nanomaterial attractive for future ultrafast photonics, starting from ultrashort pulse generation, for applications such as single-photon sensors and emitters.

## Figures and Tables

**Figure 1 nanomaterials-12-03864-f001:**
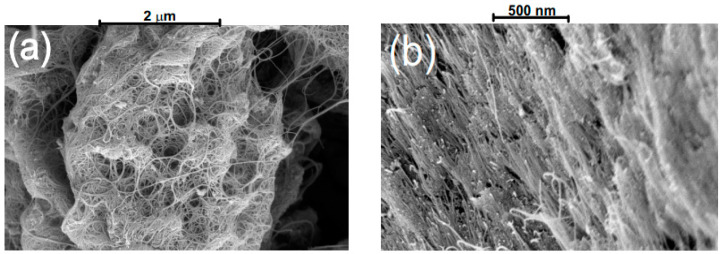
SEM images of (**a**) the initial SWCNTs, and (**b**) after high-pressure high-temperature treatment.

**Figure 2 nanomaterials-12-03864-f002:**
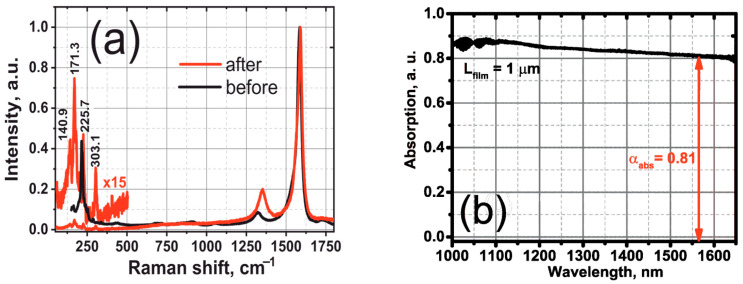
(**a**) Raman spectra of the initial SWCNTs and HDWA-SWCNTs; (**b**) a small signal absorption spectrum of HDWA-SWCNTs thin film.

**Figure 3 nanomaterials-12-03864-f003:**
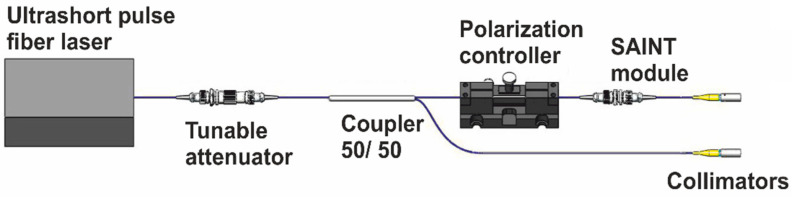
Setup for experimental measurement of a HDWA-SWCNT thin film optical loss.

**Figure 4 nanomaterials-12-03864-f004:**
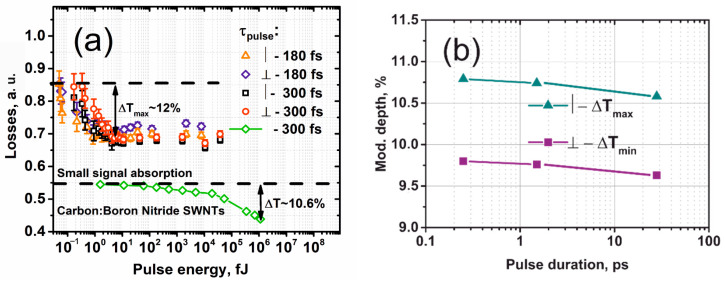
(**a**) Dependence of nonlinear optical losses of HDWA-SWCNTs and carbon nanotubes doped with boron and nitrogen using a homemade 300-fs USP source; (**b**) modulation depth Δ*T* of HDWA-SWCNTs vs. pulse width.

**Figure 5 nanomaterials-12-03864-f005:**
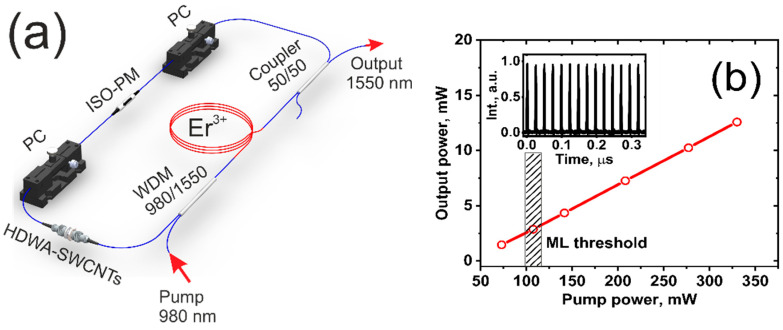
(**a**) Experimental setup of the hybrid ML fiber laser. (**b**) Output power vs. pump power and output pulse trace.

**Figure 6 nanomaterials-12-03864-f006:**
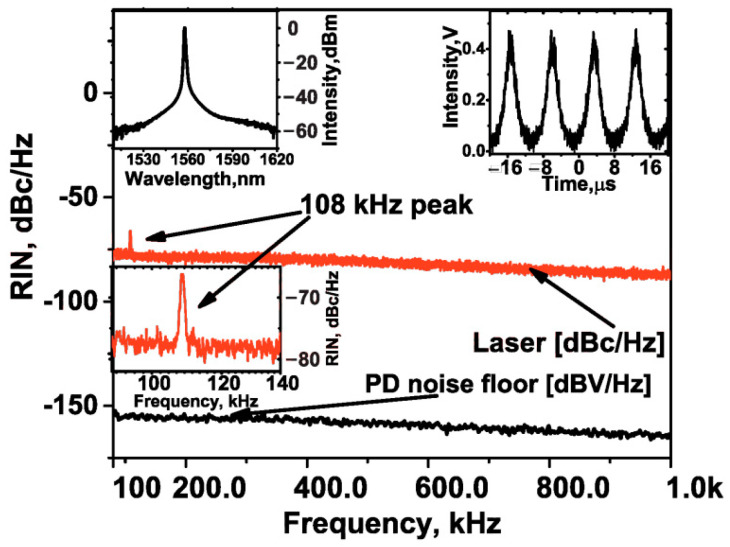
RIN of the Q-switching operation mode and photodetector (PD) + ESA noise floor. Inset: optical spectrum and output pulse trace.

**Figure 7 nanomaterials-12-03864-f007:**
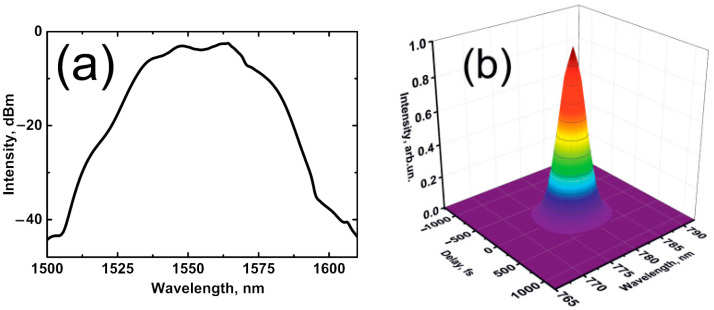
(**a**) Optical spectrum and (**b**) SHG FROG trace of the pulse profile.

**Figure 8 nanomaterials-12-03864-f008:**
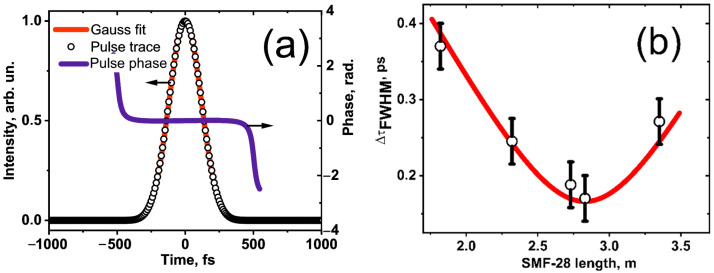
(**a**) Autocorrelation trace of USP and Gaussian fitting, along with the pulse phase. (**b**) Pulse width vs. length of output SMF-28 fiber.

**Figure 9 nanomaterials-12-03864-f009:**
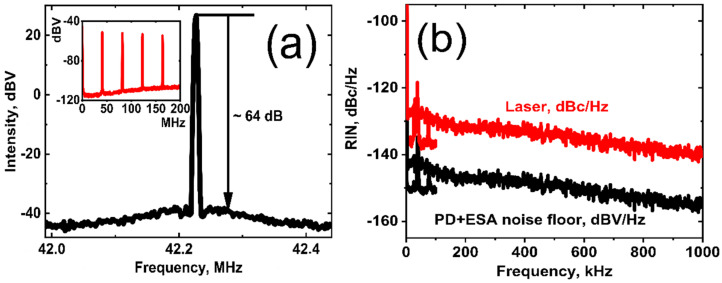
(**a**) The radio-frequency spectrum of the pulse train at the fundamental repetition frequency and in the range 30 kHz–200 MHz (inset). (**b**) RIN of the USP fiber laser and photodetector (PD) + ESA noise floor.

**Figure 10 nanomaterials-12-03864-f010:**
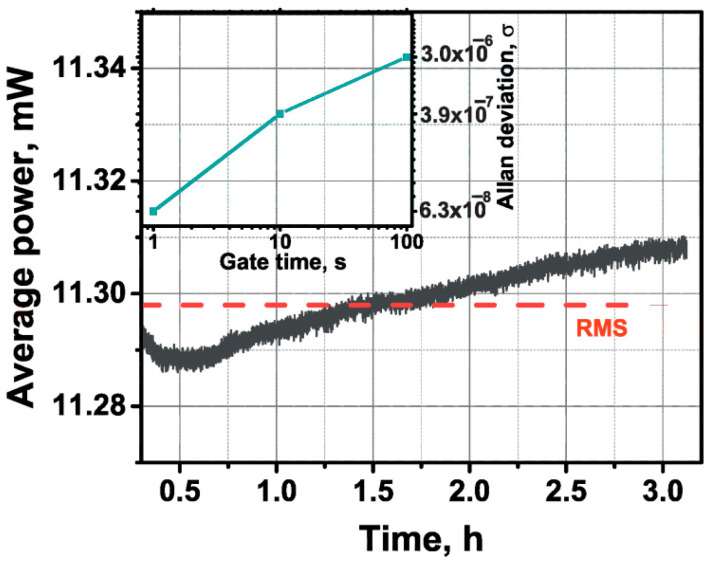
Average output of optical power during the measurement. Inset: Allan deviation of the repetition rate in a 1 × 10^2^ s time interval.

**Table 1 nanomaterials-12-03864-t001:** SWCNTs thin films parameters, along with the corresponding ML laser parameters.

Sample	Conventional SWCNTs Thin Film [28]	Carbon:Boron Nitride Single-Walled Nanotubes Thin-Film [28]	HDWA-SWCNTs Thin Film
*l_ns_*, %	18.4 ± 0.5	29.4 ± 0.3	73 ± 0.6
Δ*T*_max_, %	4.6 ± 0.5	14.9 ± 0.3	12 ± 0.6
*E_sat_*, pJ	55 ± 13	21 ± 3	<1 × 10^−3^
*l*_0_, %	33	55	85
*τ_resp_*, ps	1.06	0.50	<0.25
*P_ML_*, mW	53	85	104

## Data Availability

Not applicable.

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
