# Peer review of "Femtosecond Er-Doped All-Fiber Laser with High-Density Well-Aligned Carbon-Nanotube-Based Thin-Film Saturable Absorber"

_nanomaterials, 2022, doi:10.3390/nano12213864_

Round 1

Reviewer 1 Report

This research is very interesting and is expected to contribute to the development of lasers using nanomaterials.

Q1. When the parameter of CNT is changed, how does the result change? That is, even if the conventional SWCNT is tested by changing the length, does the difference in the results occur significantly? If so, please explain why.

Q2. How about dividing the graph in Figure 4? A lot of information is contained in one graph, making it difficult to understand at once.

Q3. In Figure 10, time is plotted on the x-axis. What does 0,56 mean? Is it 0.56?

Author Response

Thank you very much for reading our article in detail. I hope our answers to your questions will be comprehensive for you.

Q1. When the parameter of CNT is changed, how does the result change? That is, even if the conventional SWCNT is tested by changing the length, does the difference in the results occur significantly? If so, please explain why.

A1. Based on theoretical concepts, the team of authors can only assume that in the case of densification of such structures, the parameters that determine the efficiency of operation in USP lasers will be the distributions of the density and diameters of nanotubes in the sample. It is very likely that in traditional nanotubes the key parameters are the same. However, unfortunately, in this work, we did not study the influence of the size parameters of nanotubes on the generation of ultrashort pulses.

Q2. How about dividing the graph in Figure 4? A lot of information is contained in one graph, making it difficult to understand at once.

A2. We made edits in the drawings in accordance with the remarks of the reviewer.

Q3. In Figure 10, time is plotted on the x-axis. What does 0,56 mean? Is it 0.56?

The authors thank the reviewer for the remark. Indeed, the schedule lacked the standard for the references. We made corrections.

Reviewer 2 Report

In this manuscript, the authors proposed a femtosecond Er-doped all-fiber laser with hybrid mode-locked Er-doped fiber laser with high-density well-aligned carbon-nanotube-based thin-film saturable absorber. The theoretical analysis and experimental results show that the proposed model and experimental test are feasible. However, there are some points should be emphasized and interpreted.

1. The innovation points of the paper need to be further refined.

2.First abbreviations should be given full names, such as SA, NPE.

3. The data in Figure 1 is inconsistent with the description this manuscript. Please check.

4. Why use polarization maintaining polarization correlation isolator? Can ordinary isolators be used? Please give reasons.

5. Some figures are not standardized, such as the text size in the figures, such as Figure3 and Figure10. Check the vertical coordinate value in Figure 2.

6. Authors should carefully check the manuscript before submission, such as tense.

Author Response

The team of authors is grateful to you for a detailed study of our article. I hope our answers to your questions will be exhaustive for you.

Q1. The innovation points of the paper need to be further refined.

A1. We complemented the novelty associated with a low threshold for self -start of mode-locking in a hybrid scheme with high -dense carbon nanotubes.

Q2. First abbreviations should be given full names, such as SA, NPE.

A2. We made edits in accordance with the remarks of the reviewer.

Q3. The data in Figure 1 is inconsistent with the description this manuscript. Please check.

A3. Corrected

Q4. Why use polarization maintaining polarization correlation isolator? Can ordinary isolators be used? Please give reasons.

A4. The polarization maintaining isolator is used for two things. Firstly, it blocks the back propagation of radiation in the resonator. Secondly, it weakens the orthogonal polarization in the ellipse, making the radiation linearly polarized. In the process of pulse propagation, it is assumed that the state of the laser resonator changes nonlinearly under the action of phase and cross modulations, which arise due to the nonlinearity of the fibers and create a phase shift between orthogonally polarized components. This leads to a rotation of the polarization ellipse, but not to a change in its shape. Thus, upon multiple passage through the insulator-polarizer, the low-intensity polarization component is so weakened that this allows the mode-locking process to be triggered. Which would be impossible without the use of a polarizer in the resonator. We made appropriate amendments to the text of the article.

Q5. Some figures are not standardized, such as the text size in the figures, such as Figure3 and Figure10. Check the vertical coordinate value in Figure 2.

A5. We made edits in the figures in accordance with the remarks of the reviewer.

Q6. Authors should carefully check the manuscript before submission, such as tense.

A6. The authors thank the reviewer for his comments. Unfortunately, when submitting articles, typos inevitably happen, but we will make efforts to control the submitted text and drawings.

Reviewer 3 Report

Authors present a paper entitled “Femtosecond Er-doped all-fiber laser with  high-density well-aligned carbon-nanotube-based thin-film saturable absorber.” While the paper has some interesting aspects, I do not feel that it is appropriate for publication in its current form. I feel that some works would need to be done in order to make it more reasonable and publishable.

1.The author mentioned that several technologies have been developed to synthesize ordered SWCNT. What are the advantages of the high-temperature and high-pressure method used in this paper compared with those technologies? Can the same materials be obtained by using other technologies? Authors should give more comments on this other than the advantages of SWCNT or ordered SWCNT, which is so redundant.

2. They emphasize that the well-aligned SWCNT can control the polarization dependence of the SA and build lasers with well-defined characteristics. However, the causal relationship among them has not been stated clearly. Also, results are not enough to prove these (How to control? Control what? Affect what?).  

3. I am still confused that the mode-locking can be obtained with low threshold under so high insertion loss, 50% output, and isolator-polarizer? These sound little inconceivable.  

4. In line 169-177, authors state that “a low ML threshold compared with NPE-based only ML is attributable to the applied hybrid ML technique – the slow SA represented by HDWA-SWCNTs SA reliably triggers ML at a low pump power, while pulse shortening is caused by the NPE effect.” So, the only advantage of the introduction of HDWA-SWCNTs in the hybrid ML is low ML threshold? However, this key innovation point has not been stated in your paper (like abstract, introduction and conclusion)

5. The pulse parameters of your hybrid mode-locking lasers can be obtained easily in a only NPE based mode-locking laser (like threshold of 100 mW, pulse energy of 0.3 nJ, et. al). So, what is the truly advantage of your hybrid mode-locking lasers with HDWA-SWCNTs?

Author Response

The team of authors is very grateful to you for the review of our article. We hope that our responses to your questions will be exhaustive and comprehensive.

Point 1:  The author mentioned that several technologies have been developed to synthesize ordered SWCNT. What are the advantages of the high-temperature and high-pressure method used in this paper compared with those technologies? Can the same materials be obtained by using other technologies? Authors should give more comments on this other than the advantages of SWCNT or ordered SWCNT, which is so redundant.

Response 1:  Indeed, today, there are several methods for obtaining ordered structures of carbon nanotubes. It should be noted that basically all these methods are aimed at obtaining monolayers, which can be very in demand for high -speed electronics, but it is not required in production, for example, saturated absorbers. Moreover, the method proposed in our article is much easier, cheaper and allows you to get this material in relatively large quantities. We made appropriate amendments to the introduction of the article. We added a sentence for explanation of advantages compared to other methods in the text.

Point 2: They emphasize that the well-aligned SWCNT can control the polarization dependence of the SA and build lasers with well-defined characteristics. However, the causal relationship among them has not been stated clearly. Also, results are not enough to prove these (How to control? Control what? Affect what?). 

Response 2: The authors thank the reviewer for the exact remark. Apparently, this expression was not very correctly formulated. Here we meant that the use of ordered nanostructures with polarization dependence, processed by the same method, will increase the likelihood of obtaining repeated results when generating ultra -short pulses. Since the material will turn out to be more standardized, compared with the random structures of traditional carbon nanotubes disordered. We made corresponding changes to the text.

Point 3: I am still confused that the mode-locking can be obtained with low threshold under so high insertion loss, 50% output, and isolator-polarizer? These sound little inconceivable. 

Response 3: This is one of the advantages of single-walled high-density carbon nanotubes. They have a low saturation energy (< 1 fJ) and fast relaxation time (~250 fs), which allows them to effectively filter low-intensity fluctuations to launch mode-locking.

Point 4: In line 169-177, authors state that “a low ML threshold compared with NPE-based only ML is attributable to the applied hybrid ML technique – the slow SA represented by HDWA-SWCNTs SA reliably triggers ML at a low pump power, while pulse shortening is caused by the NPE effect.” So, the only advantage of the introduction of HDWA-SWCNTs in the hybrid ML is low ML threshold? However, this key innovation point has not been stated in your paper (like abstract, introduction and conclusion)

Response 4: In addition, the introduction of the HDWA-SWCNT into the resonator as a slow saturated absorber allows us to simplify the launch of mode- locking and get lower intensity noises. We added information about this in conclusion.

Point 5: The pulse parameters of your hybrid mode-locking lasers can be obtained easily in an only NPE based mode-locking laser (like threshold of 100 mW, pulse energy of 0.3 nJ, et. al). So, what is the truly advantage of your hybrid mode-locking lasers with HDWA-SWCNTs?

Response 5: As shown in [Kim, Jungwon and Youjian Song. “Ultralow-noise mode-locked fiber lasers and frequency combs: principles, status, and applications.” Advances in Optics and Photonics 8 (2016): 465-540.] mode-locked schemes based on natural saturable absorbers provide lower intensity noise. On the other hand, the use of NEP in the scheme makes it possible to obtain shorter pulses, due to the fast response time of this mode-locking mechanism (about 5 fs). At the same time, the advantage of high-density nanotubes is their higher radiation resistance, which allows them to work longer in the laser cavity without degradation of characteristics, as well as a relatively large modulation depth, which provides better differentiation of radiation fluctuations when mode-locking is launching. We added information about this in conclusion.

Round 2

Reviewer 2 Report

In this manuscript, a femtosecond Er-doped all-fiber laser with hybrid mode-locked Er-doped fiber laser with high-density well-aligned carbon-nanotube-based thin-film saturable absorber is proposed. The authors revise carefully the manuscript according to the opinions of experts. This manuscript is suitable for the scope of Nanomaterials and can be accepted.

Reviewer 3 Report

All my concerns have been done